# Botulinum Neurotoxin A-Induced Muscle Morphology Changes in Children with Cerebral Palsy: A One-Year Follow-Up Study

**DOI:** 10.3390/toxins17070327

**Published:** 2025-06-27

**Authors:** Charlotte Lambrechts, Nathalie De Beukelaer, Ines Vandekerckhove, Ineke Verreydt, Anke Andries, Francesco Cenni, Ghislaine Gayan-Ramirez, Kaat Desloovere, Anja Van Campenhout

**Affiliations:** 1Department of Rehabilitation Sciences, Neurorehabilitation Research Group, KU Leuven, 3000 Leuven, Belgium; nathalie.debeukelaer@kuleuven.be (N.D.B.); ines.vandekerckhove@kuleuven.be (I.V.); ineke.verreydt@kuleuven.be (I.V.); kaat.desloovere@kuleuven.be (K.D.); 2Kinesiology Laboratory, Geneva University Hospitals and Geneva University, 1205 Geneva, Switzerland; 3Department of Chronic Diseases and Metabolism, Laboratory of Respiratory Diseases and Thoracic Surgery, KU Leuven, 3000 Leuven, Belgium; anke.andries@kuleuven.be (A.A.); ghislaine.gayan-ramirez@kuleuven.be (G.G.-R.); 4Department of Clinical and Experimental Sciences, University of Brescia, 25121 Brescia, Italy; francesco.cenni@gmail.com; 5Clinical Motion Analysis Laboratory, University Hospitals Leuven, 3212 Pellenberg, Belgium; 6Department of Development and Regeneration, KU Leuven, 3000 Leuven, Belgium; anja.vancampenhout@uzleuven.be; 7Department of Orthopaedic Surgery, University Hospitals Leuven, 3000 Leuven, Belgium

**Keywords:** botulinum neurotoxin type A, cerebral palsy, muscle growth, muscle imaging

## Abstract

Botulinum neurotoxin type A (BoNT-A) is widely used to reduce spasticity in children with cerebral palsy. Despite its therapeutic benefits, incomplete muscle recovery has been observed post-treatment. This study evaluated longitudinal BoNT-A effects on muscle morphology over one year in children with CP (*n* = 26, mean age: 5.19 years ± 3.26). Three-dimensional freehand ultrasound assessed medial gastrocnemius muscle volume (MV), muscle belly length (ML), cross-sectional area (CSA), and echo intensity (EI) at baseline and at 3, 6, and 12 months post-BoNT-A. Z-score normalization accounted for natural muscle growth. Linear mixed models analyzed muscular changes over time, and repeated-measures ANOVA compared muscle parameters to an age- and severity-matched control group (n = 26, mean age: 4.98 ± 2.15) at one-year follow-up. MV exhibited a declining trend at 3 (*p* = 0.005), 6 (*p* = 0.003), and 12 months (*p* = 0.007), while ML remained unchanged throughout follow-up (*p* = 0.95). The initially reduced CSA at 6 months (*p* = 0.0005) recovered at one year, and EI increased only at 3 months post-BoNT-A (*p* < 0.0001). At one-year follow-up, there was a trend for reduced growth rate (MV/month) (*p* = 0.035) in the intervention group, whereas the control group exhibited an increased muscle growth (*p* = 0.029). These findings suggest distinct recovery timelines for CSA and ML, which may explain the incomplete MV recovery and highlight substantial interindividual variation in recovery processes.

## 1. Introduction

It is widely recognized that children with cerebral palsy present with motor impairments, in which spasticity, together with other contributing factors, plays a prominent role. Spasticity arises from a disrupted connection between the brain and muscles, resulting in enhanced muscular stretch reflexes [1]. Approximately 80% of children with cerebral palsy present with spasticity, making spastic cerebral palsy (SCP) the predominant subtype [2].

A frequently applied treatment for targeting hyperactive stretch reflexes is the intramuscular administration of botulinum neurotoxin type A (BoNT-A). By inhibiting acetylcholine release at the neuromuscular junction, BoNT-A induces focal and temporary muscle paralysis, lasting for approximately 3 months [1]. The medial gastrocnemius (MG) muscle is commonly targeted with this treatment due to its susceptibility to spasticity, which significantly impacts gross motor function [3]. When combined with adjunctive therapies, such as serial casting, orthotic management, and intensive physiotherapy, BoNT-A has been shown to reduce spasticity, increase the joint range of motion (ROM), and delay the need for orthopedic surgery [4,5,6,7,8,9].

Despite its clinical benefits, concerns regarding the impact of BoNT-A on muscle integrity have emerged. These concerns stem largely from preclinical animal studies, which have reported muscle atrophy, reduced muscle mass, and loss of contractile tissue following BoNT-A administration [10,11,12]. While these studies offer useful insights, high doses and short injection intervals are often used that do not reflect the standard clinical practice. Moreover, animal models do not fully replicate the complex neuromuscular pathology of SCP, limiting the generalizability of their findings. This underlines the need for (human) studies that are in agreement with the current clinical practice to better understand the impact of BoNT-A on muscle growth in SCP. This is particularly important given that these children already exhibit macroscopic muscle morphology deficits from an early age, including reduced muscle volume (MV), decreased cross-sectional area (CSA), and shorter muscle length (ML) [13].

In recent years, longitudinal imaging studies have provided growing evidence that intramuscular BoNT-A injections temporarily affect muscle morphology in children with SCP [14,15,16,17,18]. During the acute post-injection phase—within the first three months—reductions in MG muscle volume, normalized to skeletal growth (nMV), have been observed [14,15,16]. These changes occurred regardless of whether children were receiving BoNT-A for the first time or had a history of prior treatment. Notably, echo intensity values (EI), as indirect indications of muscle quality, remained unaffected within the acute phase post-BoNT-A treatment [16].

Following this acute phase, the neurotoxic effects of BoNT-A gradually subside as reinnervation begins, marking the onset of a recovery phase, which lasts up to six months [19]. However, evidence suggests that muscle recovery during this period may be incomplete. Reductions in nMV and normalized cross-sectional area (nCSA) have been noted, while normalized muscle length (nML) remains relatively stable [15,17]. An upward trend in EI values following BoNT-A treatment was observed in both previously untreated and previously treated children [17].

To date, the long-term impact of BoNT-A on muscle growth beyond six months remains largely unexplored. One study indicated that muscle size increased relative to baseline, but muscle growth rates were reduced compared to pre-injection rates, which was in contrast with the observed growth rate increase in typically developing (TD) children [20]. Due to the absence of an untreated SCP control group, it remains unclear whether these growth rate differences resulted from BoNT-A administration or reflected the inherent growth deficits in children with SCP. Furthermore, muscle size was not normalized to skeletal growth, thus not excluding natural muscle growth caused by increases in body size dimensions of the child. Consequently, the observed muscle size increases after BoNT-A may have reflected typical muscle growth due to anthropometric growth of the child, and the true pathological effect of BoNT-A on muscle growth at one year follow-up remained unknown.

In summary, the long-term effects of BoNT-A injections on muscle growth remain poorly understood. Longitudinal research with multiple timepoints after BoNT-A within the same cohort is needed to better understand post-BoNT-A muscle growth evolution.

Therefore, the primary aim of this study was to investigate the effect of BoNT-A on MG (morphology) parameters in children with SCP (intervention group) at one year post-injection, with intermediate assessments at 3 and 6 months using three-dimensional freehand ultrasound (3DfUS). Additionally, we explored the potential influence of BoNT-A history and disease severity on muscle recovery. The second aim was to compare the muscle (morphology) parameters between baseline and only at the one-year follow-up of the intervention group, with an age- and severity-matched longitudinal control group of children with SCP who did not receive BoNT-A injections during the follow-up period. In addition, we qualitatively compared the one-year morphological muscle growth of the intervention group to an age-matched cross-sectional SCP reference database to contextualize findings relative to a historical SCP cohort. We hypothesized that muscle size recovery in the intervention group would be completed one year post-BoNT-A. For the second aim, we expected no change in muscle morphology parameters at one-year follow-up of the intervention group compared to the control group.

## 2. Results

### 2.1. Participant Characteristics

Twenty-six children aged 2.8–9.2 years who had been scheduled for BoNT-A administration were included in the intervention group. A visual representation of the number of measurements per participant is displayed in Appendix A. All children had a baseline measurement (i.e., prior to the BoNT-A injection) and a one-year post-BoNT-A follow-up measurement, and 17 children were also measured at 3 and 6 months post-BoNT-A. Of the 26 participants, six had only one intermediate measurement at 3 months (n = 3) or at 6 months (n = 3), while 3 children had no intermediate measurements. Patient characteristics at all timepoints can be found in Appendix A. The average time between the baseline 3DfUS measurement and the BoNT-A administration was 14.8 days (range, 0–70 days), and the average onabotulinum toxin type A (Botox, Allergan) dose for the MG was 2.41 (standard deviation (SD), 0.87) units/kg body weight. The age- and disease severity-matched control group consisted of 26 children who had not received BoNT-A injections in the year prior to the baseline measurement and were not planned for BoNT-A during the follow-up period. All children of this control group were assessed at 2 timepoints with a one-year follow-up interval. Additionally, a cross-sectional SCP reference database including 87 children with no BoNT-A administration in the MG during the year before the measurement was used as a reference to qualitatively describe the muscle growth of the intervention group from baseline to one-year follow-up. The baseline patient characteristics of all groups can be found in Table 1.

### 2.2. Muscle Morphology and Quality Following BoNT-A Administration

The results of the muscle morphology outcomes (MV, ML, and CSA) are presented as z-scores (zMV, zML, zCSA), which are based on anthropometry-related percentile curves of TD children, to exclude the impact of natural growth during follow-up and to address the lack of precision of ratio scaling when applied across wide age ranges [13]. This approach allows for standardized comparisons within and between groups and outcomes [21]. These z-scores are further labeled as “deficits”.

The results of linear mixed-models (LMM) analyses showed an effect of time for zMV (F = 4.15, *p* = 0.001, η^2^p = 0.17), zCSA (F = 5.08, *p* = 0.003, η^2^p = 0.19), and EI (F = 6.77, *p* = 0.0005, η^2^p = 0.24), but not for zML (F = 0.12, *p* = 0.95, η^2^p = 0.014). Pairwise comparisons revealed a trend for an increased MV deficit at 3 months (*p* = 0.005), 6 months (*p* = 0.003), and 1 year (*p* = 0.007) compared to baseline (Table 2 and Figure 1). The CSA deficit significantly increased between baseline and 6 months (*p* = 0.0005), but recovered again between 6 months and 1 year post-BoNT-A (*p* = 0.003) (Table 2 and Figure 1). Muscle length did not change over time. EI showed a significant increase at 3 months post-BoNT-A (*p* < 0.0001) but recovered at 6 months (Table 2 and Figure 1). Figure 2 presents individual trajectories for zMV, zCSA, and zML per GMFCS level. Appendix A, provides an overview of the estimates per timepoint for the muscle (morphology) parameters.

In addition to z-scores, muscle morphology parameters were also expressed as normalized values using ratio scaling to facilitate comparison with the previous literature. The results of LMMs for the normalized muscle morphology outcomes can be found in Appendix A, and estimates per timepoint are presented in Appendix A.

#### 2.2.1. Effect of Disease Severity on Muscle Recovery After BoNT-A Injection

No differences between the GMFCS I and GMFCS II + III groups were present at baseline (Appendix A), except for a trend for a larger deficit of MV (+0.53 z-score, *p* = 0.031) and ML (+1.35 z-score, *p* = 0.058) in the GMFCS II + III group compared to the GMFCS I group. In Appendix A, an overview of the estimates per timepoint can be found per GMFCS level for the muscle parameters expressed as z-scores, as well as for the normalized values.

LMM analyses revealed no significant interaction effect of time × group for zMV (F = 1.96, *p* = 0.13, η^2^p = 0.089), zCSA (F = 0.82, *p* = 0.49, η^2^p = 0.039), zML (F = 0.75, *p* = 0.53, η^2^p = 0.085), and EI (F = 0.41, *p* = 0.75, η^2^p = 0.020). Pairwise comparisons (Table 3) showed trends for a decreased MV at 6 months (*p* = 0.007) and 1 year (*p* = 0.006) post-BoNT-A in the GMFCS II + III group, as well as a significant increase in EI at 3 months (*p* < 0.0001). The GMFCS I group only demonstrated a trend toward a decreased CSA at a 6-month follow-up (*p* = 0.004), which recovered between 6 months and 1 year (*p* = 0.002). The results for the normalized muscle morphology outcomes can be found in Appendix A.

#### 2.2.2. Effect of BoNT-A History on Muscle Recovery After BoNT-A Injection

At the time of the baseline measurement, children with a history of BoNT-A were older (+42%, *p* = 0.025) and presented with larger body size dimensions (height (+12%, *p* = 0.016) and body weight (+24%, *p* = 0.008)) (Appendix A). There were no differences in muscle (morphology) deficits at baseline between the BoNT-A history and BoNT-A-naive groups. Estimates per timepoint can be found in Appendix A for the z-scores and normalized muscle parameters.

No significant timexgroup effect was found for zMV (F = 1.39, *p* = 0.26, η^2^p = 0.065), zCSA (F = 2.42, *p* = 0.075, η^2^p = 0.11), and EI (F = 1.56, *p* = 0.21, η^2^p = 0.072), while for zML, there was a trend towards a different recovery trajectory between both groups (F = 3.12, *p* = 0.045, η^2^p = 0.28). Pairwise comparisons (Table 4) showed a trend for a decreased MV at 3 months (*p* = 0.004), followed by a significant MV reduction at 6 months (*p* < 0.0001) and 1 year (*p* = 0.002) post-treatment in the BoNT-A-naive group. The CSA was only significantly lower at 6 months (*p* = 0.0009), and EI values were only increased at 3 months (*p* = 0.002). The BoNT-A history group exhibited no trends or significant changes in muscle parameters at the follow-up measurements compared to baseline, with the exception of a trend for higher EI values at 3 months post-BoNT-A (*p* = 0.012). The results for the normalized muscle morphology outcomes can be found in Appendix A.

### 2.3. Muscle Growth Comparison Between the Intervention and Control Groups

No statistical differences were present between the intervention and control groups for baseline anthropometry and muscle (morphology) parameters (Appendix A.

The repeated-measures ANOVA revealed no main effect of time for zMV (F = 2.50, *p* = 0.12, η^2^p = 0.048), but there was a trend for a timexgroup effect (F = 5.90, *p* = 0.019, η^2^p = 0.11). For zCSA, both the main time and timexgroup effects were not significant (F = 1.87, *p* = 0.18, η^2^p = 0.036 and F = 1.75, *p* = 0.19, η^2^p = 0.034, respectively). Similarly, there were no significant main time and timexgroup effects for zML (F = 0.009, *p* = 0.93, η^2^p = 0.000 and F = 0.017, *p* = 0.90, η^2^p = 0.000, respectively). The overall time effect for EI showed no significance (F = 0.056, *p* = 0.81, η^2^p = 0.001), but there was a trend for a timexgroup effect (F = 6.013, *p* = 0.018, η^2^p = 0.107). There was no overall effect of time for growth rate (i.e., the increase in muscle volume per month) (F = 0.004, *p* = 0.95, η^2^p = 0.000), while the timexgroup effect was significant (F = 9.78, *p* = 0.003, η^2^p = 0.16). Individual datapoints of difference scores between baseline and follow-up for each group are displayed in Figure 3. Post hoc analyses (Table 5) revealed a significant increase in MV deficit (*p* = 0.007) and a significant reduction in growth rate (*p* = 0.035) between baseline and follow-up in the intervention group, while growth rate significantly increased in the control group (*p* = 0.029). Appendix A presents the individual muscle morphology trajectories between baseline and follow-up of the intervention group together with the corresponding case–control match.

The results of the repeated-measures ANOVA for the normalized muscle size outcomes can be found in Appendix A.

### 2.4. Muscle Size Comparison Between the Intervention Group and SCP-Reference Database

Figure 4 qualitatively represents the exploration of the pre–post-BoNT-A muscle size outcomes of the intervention group relative to the SCP reference group. Individual muscle size trajectories fit within the muscle size ranges of the age-matched SCP reference database and highlight a large heterogeneity between patients, with no clear trends for the GMFCS and BoNT-A history groups.

## 3. Discussion

This longitudinal follow-up study primarily aimed to investigate the effect of BoNT-A on muscle (morphology) parameters up to one year post-injection. Additionally, the effect of disease severity and BoNT-A history on muscle size recovery was explored. The second aim was to compare the muscle outcomes at one year post-BoNT-A to an age- and severity-matched untreated control group. A qualitative description of the muscle morphology parameters at one year after injection was also made relative to an age-matched cross-sectional SCP reference database. It was hypothesized that muscle size in the intervention group would recover within a year post-BoNT-A, and we expected no changes in muscle outcomes after one year compared to the control group.

### 3.1. Time Course of Muscle (Morphology) Parameters After BoNT-A Injection

#### 3.1.1. Muscle Volume

In general, there was a trend for a decreased MV at 3 and 6 months post-BoNT-A and, to a lesser extent, at one year compared to baseline. In particular, the MV deficit increased between baseline and 3 months post-BoNT-A (+0.50 z-score, *p* = 0.005). When expressed in normalized values, MV decreased by 10% (*p* = 0.006), which is in correspondence with the previous literature that reported a decline in normalized MV of 5–11% at 8–13 weeks post-BoNT-A in children with SCP [15,16]. Previously reported inter-rater–inter-session reliability data for nMV in children with SCP indicated a standard error of measurement (SEM) of 4%, reinforcing the interpretation that the observed changes in MV following BoNT-A injections are meaningful [22]. Six months post-BoNT-A, the reduction in MV was still present. There was a trend for an increased MV deficit (+0.42 z-score, *p* = 0.003), which corresponded with a reduction in normalized MV of 7% (*p* = 0.007). Similarly, Alexander et al. reported a 10% MV decrease at 6 months post-BoNT-A in children that received their first-time injection [15]. De Beukelaer et al. also reported a trend for a decreased normalized MV between baseline and 6 months in a BoNT-A-naïve group and a BoNT-A history group [17]. Although it was hypothesized that muscle size would be recovered at one year post-BoNT-A, MV was still lower compared to the baseline assessment, with a trend towards a larger MV deficit (+0.34 z-score, *p* = 0.007) after one year, while the normalized MV showed no statistical difference at one year compared to baseline (*p* = 0.023). Interestingly, the previous literature reported persistence of muscle atrophy in 2 healthy participants up to 12 months after a single dose of BoNT-A [23].

These results suggest that MV recovery is still incomplete one year after treatment, which should be taken into account when determining treatment indications. Future research is needed to investigate the impact of additional interventions, such as strength training of the injected muscle, that may help promote muscle growth [24]. However, when interpreting these findings, it is important to acknowledge that BoNT-A was used in combination with adjunct therapies such as serial casting (n = 24) and the use of ankle foot orthoses (n = 26), which may have contributed to muscle atrophy. More research is needed to understand the isolated impact of BoNT-A on muscular alterations.

#### 3.1.2. Cross-Sectional Area

Following BoNT-A administration, the CSA was only reduced at 6 months, with a significant CSA deficit increase of 0.82 z-score (*p* = 0.0005), and a significant 12% decrease in normalized CSA (*p* < 0.0001) compared to baseline. With an inter-rater–inter-session SEM of 5% for nCSA, the observed difference exceeded the threshold of measurement error, indicating a meaningful change [22]. To the best of our knowledge, only one study examined the CSA at 6 months post-BoNT-A, and reported a 10% and 17% reduction in normalized CSA in children with a history of prior BoNT-A injections and in children receiving their first BoNT-A treatment, respectively [17]. Hence, our findings corroborate the previous literature suggesting impaired muscle growth of the medial gastrocnemius CSA 6 months following BoNT-A administration. It should be noted, however, that there is an overlap (n = 12) between the children included in the latter and the current study, which contributes to the similarity of the results found. In animal studies, a reduction in fiber CSA was observed in rats at 6 months post-BoNT [12]. Conversely, Deschrevel et al. found a non-significant increase in the fiber CSA of the medial gastrocnemius in SCP children receiving their first BoNT-A injection at 6 months post-BoNT-A [25], which contradicts our results that suggest a reduction of the cross-sectional muscle dimension. This discrepancy could be explained by the weak correlation between the microscopic and macroscopic muscle size parameters [26]. The decreased CSA at 6 months was followed by a trend toward recovery, with improvements in both the CSA deficit (−0.015 z-score, *p* = 0.003) and normalized CSA (+10%, *p* = 0.002) between 6 months and one year. At one-year follow-up, no significant differences were present compared to baseline, indicating a recovery of the cross-sectional muscle dimension one year after BoNT-A treatment.

#### 3.1.3. Muscle Length

Muscle length was preserved at all timepoints. Although insignificant, the mean difference values suggest that longitudinal muscle growth may even be slightly enhanced following BoNT-A administration. Since BoNT-A primarily targets the neural component of ankle joint resistance by reducing hyperactive stretch reflexes, this reduction in spasticity may lead to an increased ROM, which results in an improved gait pattern [27], thereby facilitating greater muscle elongation. Additionally, previous research showed that BoNT-A increases the resting ML of the medial gastrocnemius at 2 weeks post-injection [28]. Furthermore, BoNT-A is often implemented within a multimodal therapeutic approach, including ankle foot orthoses, intensive physiotherapy, and serial casting [3,29,30], which may further contribute to the preservation of muscle length.

#### 3.1.4. Integration of Muscle Morphology Outcomes

In general, it should be noted that there was a trend for a reduced MV at one year follow-up, while no trends for a decreased CSA and ML were observed after one year. This discrepancy may be explained by the individual trajectories presented in Figure 2. More specifically, changes in MV are defined by a combination of cross-sectional and longitudinal changes, represented by the CSA and ML [31]. A decrease in MV can result from a reduction in the CSA (suggesting muscle atrophy) and/or a reduction in ML (suggesting the development of muscle contractures). In Figure 2, it can be observed that some children developed contractures after one year, resulting in shorter muscles, rather than having a reduced CSA. Additionally, the CSA is a measurement taken at a specific location within the muscle, namely at the midpoint of the muscle belly, while MV is estimated over the entire length of the muscle belly. This could partly explain why a change in MV may be observed, even though it is not reflected in the CSA. This reflects that the response to treatment is highly individual and highlights the need for careful follow-up of the entire muscle structure.

#### 3.1.5. Muscle Quality

Muscle quality, indirectly assessed via EI, showed a reduction in the acute phase after BoNT-A treatment. This was reflected by a 6% increase in EI at 3 months post-injection (*p* < 0.0001). Given an SEM of 3%, the observed change in EI exceeded the expected measurement error [22], suggesting a change in the ratio of contractile to non-contractile tissue [32,33]. Interestingly, a preliminary MRI study in children with SCP reported a trend toward increased fat fraction values at 3 months post-BoNT-A injection, indicating a potential short-term adverse effect of BoNT-A on muscle quality [34]. Similarly, animal studies have demonstrated a loss of contractile muscle material accompanied by fatty infiltration and increased collagen deposition at 1–6 months post-injection [10,11,35,36]. However, caution is warranted when comparing these findings, as EI cannot differentiate between fat and collagen and provides only an indirect estimate of muscle integrity. Notably, this increased EI at 3 months post-BoNT-A was not observed in a previous study investigating the short-term effects of BoNT-A in children with SCP [16]. Between 3 and 6 months, EI values returned to baseline levels, indicating the recovery of muscle quality by 6 months, which persisted up to one year post-BoNT-A. In contrast to these findings, a previous study reported a trend toward increased EI at 6 months post-injection in both BoNT-A-naive children and children with a history of prior BoNT-A administrations [17]. In summary, the literature on the effects of BoNT-A on muscle quality is conflicting, underscoring the need for studies with high-quality measurement tools, such as MRI, to accurately assess muscle quality post-BoNT-A treatment.

#### 3.1.6. Effect of Disease Severity and History of Previous BoNT-A Injections on Muscle Recovery

This study also explored the possible effect of disease severity (i.e., GMFCS level) and prior BoNT-A injection history on muscle outcomes. Overall, no significant differences were observed in the trajectory of muscle outcomes over time between the two GMFCS groups. However, some interesting differences were noted at specific timepoints. Specifically, a reduction in MV and muscle quality was mainly observed in the GMFCS II + III group. For the BoNT-A history groups, no clear differences were found between the time courses of muscle parameters, except for a trend suggesting a difference in ML progression. The mean differences suggest an increase in ML at 3 months in the BoNT-A-naive group, which subsequently decreased at 6 months but recovered by one year, approaching baseline levels. In contrast, the BoNT-A history group exhibited a reduction in ML at 3 months, followed by an increase at 6 months. At the one-year follow-up, ML slightly declined again to a level comparable to baseline. However, it should be noted that the mean differences between the different timepoints were not statistically significant. The results for MV and the CSA indicate that muscle atrophy was primarily observed in the group with no prior history of BoNT-A injections, suggesting that the first BoNT-A injection leads to the most pronounced muscle atrophy. Similarly, De Beukelaer et al. investigated muscle morphology following BoNT-A administration in children with a history of prior BoNT-A injections and children without previous exposure. Although the results did not show a statistically significant difference between the groups, the BoNT-A history group tended to exhibit greater reductions in MV and the CSA [17]. However, it is important to note that in the current study, the BoNT-A-naive group consisted of 17 patients, while the BoNT-A history group included only 9 patients. This smaller sample size in the history group may have limited the statistical power to detect significant differences in this group.

### 3.2. Muscle Comparison Between the Intervention and Control Groups at the One-Year Follow-Up

The intervention and the age- and severity-matched control groups exhibited distinct trajectories over the one-year follow-up period for MV, EI, and growth rate. After one year, the MV of the intervention group was reduced, which was reflected by an increased MV deficit of 0.34 z-score (*p* = 0.0076) and a trend for a 6% decreased normalized MV (*p* = 0.035), whereas the control group showed no significant changes over time.

A trend for different time trajectories was also observed for muscle quality (*p* = 0.018). In the intervention group, EI values were higher at one year compared to baseline, while the control group had lower echo intensity outcomes at the one-year follow-up. However, changes in EI over time were not significant neither in the intervention group (*p* = 0.12) nor in the control group (*p* = 0.063).

Growth rate trajectories were significantly different between both groups (*p* = 0.003). Specifically, growth rates in the intervention group significantly declined following BoNT-A injections (−30%, *p* = 0.035), while the control group exhibited a significant increase in growth rate over the one-year follow-up period (+32%, *p* = 0.029). Compared to the previous literature, the current post-BoNT-A growth rate of the intervention group (0.34 mL/month) exceeded the growth rates reported at 2–3 months (median growth rate of 0.00 mL/month) [16] and 6 months in BoNT-A-naive children (median growth rate of 0.038 mL/month) and in children with a history of BoNT-A injection (median growth rate of 0.21 mL/month) [17] post-BoNT-A, which is suggestive of a more favorable muscle growth recovery over time. Barber et al. reported a similar mean growth rate of 0.31–0.36 mL/month at 12 months post-BoNT-A, which was significantly lower compared to the growth rate of the control group consisting of TD children. However, no control group consisting of children with SCP that did not receive BoNT-A was included, making it unclear whether the reduced growth rate was attributable to BoNT-A injections or to the underlying mechanisms of CP. In the present study, each child in the intervention group was case–control-matched with a child with SCP who had not received BoNT-A injection in the MG in the past year. Case–control matching for disease severity and age was challenging, and matching on baseline muscle deficits was not feasible, as illustrated in Appendix A. This demonstrates that children who were case–control-matched based on age and disease severity varied significantly in baseline muscle deficits. A randomized controlled trial would be the optimal study design to address this problem, but ethical constraints make such a study difficult to conduct.

Overall, our findings suggest that the children with CP who had received BoNT-A exhibited less favorable muscle growth during the year following BoNT-A treatment, which was evident in a significant difference in growth rate evolution from baseline to the one-year follow-up between the intervention and control groups. Recent research has demonstrated that BoNT-A injections influence the expression of proteins involved in muscle growth, potentially disrupting the key signaling pathways essential for muscle development [37]. Furthermore, in vitro studies suggest that BoNT-A can reduce the fusion index of muscle cells, indicating a negative impact on muscle repair and growth [38].

### 3.3. Qualitative Muscle Size Description Post-BoNT-A Relative to the SCP Reference Database

The qualitative analysis of muscle size outcomes, presented in Figure 4, indicates that muscle size after one year of BoNT-A administration was comparable to the muscle size of the age-matched cross-sectional SCP reference database. This is consistent with findings on muscle size following other spasticity-reducing interventions, such as selective dorsal rhizotomy [39]. Overall, individual trajectories varied considerably between the patients, with no clear trends observed across the GMFCS level groups or the BoNT-A history groups. The large heterogeneity in muscle recovery between the patients underscores the need for patient-tailored treatment plans following BoNT-A injections. Individualized follow-up is essential for monitoring morphological muscle changes over time. Timely implementation of supplementary interventions during the recovery phase post-BoNT-A, including targeted strength training of the injected muscle [24], potentially combined with nutritional supplementation [40,41,42], may support optimal muscle recovery. However, further research on interventions targeting muscle size recovery is needed.

### 3.4. Summary

Although MG muscle morphology and quality are affected by BoNT-A, a clear regenerative process is evident within the year following treatment. It should also be acknowledged that spastic muscles inherently exhibit structural abnormalities, including reduced muscle size and increased amounts of connective tissue. These alterations can impact the ultrasound appearance of the muscle and may influence both the BoNT-A injection strategy and the muscle’s recovery following treatment [43]. Despite these considerations, the present study highlights the potential of 3DfUS as a non-invasive and clinically applicable technique capable of detecting longitudinal changes in muscle morphology and quality. This offers promising opportunities for improving the monitoring and personalization of care for children with spasticity.

## 4. Conclusions

In line with the previous literature on short- and medium-term effects post-BoNT-A, a trend for incomplete muscle size recovery was observed in the long term (i.e., one year after BoNT-A treatment), primarily affecting muscle volume. While longitudinal muscle growth remained stable throughout the entire follow-up period, cross-sectional muscle growth showed a decline at six months post-BoNT-A, but entirely recovered at 12 months post-BoNT. Muscle quality initially decreased significantly at 3 months, but subsequently recovered in the following months.

This was the first study to incorporate a case–control-matched group for age and disease severity, revealing distinct muscle growth trajectories over a one-year period between the groups. Specifically, muscle volume decreased one year post-BoNT-A in the intervention group, whereas it remained stable in the control group. Similarly, the growth rate declined in the intervention group during the year following BoNT-A injection, while the control group exhibited a significant increase.

Furthermore, the results revealed considerable inter-individual variability in responses post-BoNT-A, emphasizing the need for personalized, long-term monitoring.

## 5. Materials and Methods

### 5.1. Participants

For this longitudinal study, participants were recruited via the Cerebral Palsy Reference Centre and Clinical Motion Analysis Laboratory of the University Hospitals Leuven (Belgium) as part of three parallel research projects. Patients who were clinically planned for BoNT-A administration in the MG were eligible if they met the following criteria: (1) diagnosis of SCP, confirmed by a neuro-pediatrician; (2) aged between 2 and 9 years at the moment of inclusion; (3) Gross Motor Function Classification System (GMFCS) level of I–III; (4) uni- or bilateral involvement; (5) no history of previous BoNT-A injections in the MG in the previous 12 months; (6) no history of orthopedic surgery in the previous 2 years; (7) no surgery in the targeted MG; and (8) no severe comorbidities (e.g., cognitive problems). The same inclusion criteria were applied to the SCP control group where the patients were not planned for BoNT-A injections. These children were case–control-matched to the BoNT-A group according to the baseline age and GMFCS level. If multiple control children were eligible, the best match was selected based on the corresponding topographical involvement (unilateral/bilateral). Additionally, a cross-sectional SCP reference database was included to qualitatively describe the muscle morphology over one year post-BoNT-A of the intervention group relative to the SCP reference data. The included SCP children in the reference database were aged between 2–10 years and met the remaining inclusion criteria as stated above, and thus did not receive BoNT-A injections in the past 12 months prior to the muscle morphology assessments. This study was approved by the Ethics Committee of UZ/KU Leuven (S59945, S62187, S62645) under the Declaration of Helsinki. Written informed consent was obtained for all the children from their parents or legal guardians.

### 5.2. BoNT-A Administration

Guided by ultrasonography and under general anesthesia, BoNT-A (Botox^®^, Allergan, Diegem, Belgium) injections were applied to the MG by a pediatric orthopedic surgeon. If clinically relevant, other muscles were also injected. The patient-specific BoNT-A dose was determined based on a clinical examination (spasticity and ROM assessments) and 3D gait analysis (to evaluate the involvement of the spastic MG on the gait pattern) prior to the administration. As part of the standard clinical care, serial casting (if necessary), orthotic management, and intensive physiotherapy were applied as adjunctive treatment after the BoNT-A administration. This integrated approach ensures re-injection intervals of at least one year, when clinically indicated [44].

### 5.3. Three-Dimensional Freehand Ultrasound Assessment

To quantify the macroscopic muscle morphology of the MG, a valid and reliable 3DfUS protocol was applied [45]. The inter-rater–inter-session reliability of the MG, both absolute (SEM and SEM%) and relative (intraclass correlation coefficient), was rated as good to excellent [22]. Prior to the 3DfUS assessment, anthropometric data including body weight and height were registered at baseline and at each follow-up measurement. The most involved leg of the children was selected based on spasticity outcomes of the gastrocnemius (MAS together with the Tardieu R1 angle) and the maximal ankle dorsiflexion ROM, which was measured with a goniometer while the knee was extended.

During the 3DfUS assessment, the participants were placed in the prone position with their lower leg supported on a triangle cushion, while the foot was hanging over the edge of the cushion to ensure a resting position of the ankle (Figure 5). Data collection and processing of the MG morphology were performed using STRADWIN software (version 6.0; Department of Engineering, Cambridge University, Cambridge, UK). A 3D reconstruction of the MG was completed by combining a conventional two-dimensional (2D) US (Telemed-Echoblaster B-mode ultrasound device, Telemed Ltd., Vilnius, Lithuania) with motion tracking (Optitrack V120:Trio, NaturalPoint Inc., Corvallis, OR, USA) of the reflective markers attached on the US probe. In this way, the orientation and translation of each 2D US image was determined while sweeping over the muscle [45]. Large quantities of an acoustic transmission gel were applied in conjunction with the Portico, a custom-designed plastic mount integrated with a gel pad, to reduce muscle deformation during the ultrasound measurement [46]. US settings were held constant during the acquisition and across timepoints. From the acquired 3D image, muscle morphology parameters including MV (in milliliters, ml), ML (in milimeters, mm), CSA (in squared milimeters, mm^2^), and indirect muscle quality parameter EI (gray scale, 0–265 bit), were extracted. MV was defined as the measure of the 3D volume of the MG muscle belly calculated through the cubic planimetry technique [47]. ML referred to the distance between the most superficial point of the medial femoral condyle and the most proximal point of the muscle–tendon junction. The anatomical CSA represented the 2D cross-sectional area at 50% of the muscle belly length, oriented perpendicularly to the muscle’s longitudinal axis. Muscle quality was estimated by calculating the average EI value across the entire 3D reconstruction, with higher values representing worse muscle quality. The 3DfUS measurements and data processing were performed by multiple researchers with a minimum of one year of experience.

Muscle size parameters (MV, CSA and ML) were expressed in z-scores, which are derived from the anthropometric-related percentile curves from TD children (n = 145) [21]. These unitless z-scores reflect alterations as deficits independent of anthropometric growth and serve as the primary outcome measure. To allow comparison with other studies, normalized muscle size outcomes using ratio scaling [13] will be reported as secondary outcomes. Specifically, MV (in mL), CSA (in mm^2^), and ML (in mm) were scaled relative to body weight (kg)xheight (m), body weight (kg), and height (m).

### 5.4. Statistics

Normality was checked using the Shapiro–Wilk test, and outliers were identified through visual inspection of boxplots. Since most parameters were normally distributed, data were presented as the means (standard deviations) or as estimates (confidence intervals (lower bounds, upper bounds)), unless stated otherwise. The significant alpha level was set at 0.013 after Bonferroni correction for four muscle (morphology) parameters (MV, CSA, ML, EI).

To investigate the first aim of this study (i.e., examining the effect of time on muscle (morphology) outcomes following BoNT-A administration), linear mixed models were applied with time as a fixed effect. This statistical approach was chosen to account for missing data and variable time intervals. Based on the likelihood ratio test, either a random intercept model or a repeated model with an unstructured covariance matrix was selected for each muscle parameter. Bonferroni correction was applied to account for six pairwise comparisons (0.013/6 = 0.002). Additionally, interaction effects between GMFCS level and time and BoNT-A history and time were investigated in separate models to explore their potential influence on the muscle (morphology) outcomes. Partial eta squared (η^2^p) was derived for the main effect as an estimate of effect size, with interpretation thresholds of 0.01 (small), 0.06 (moderate), and 0.14 (large). Analyses were performed in SAS^®^ (version 8.3, Statistical Analysis Software, SAS Institute Inc., Cary, NC, USA).

The second aim was to compare the muscular changes between baseline and the 1-year follow-up between the intervention and control groups. First, an unpaired t-test was used to compare the baseline characteristics between the groups. Next, a repeated-measures ANOVA with both within-subject (time) and between-subject (group) factors was conducted to examine the main effect of time and the timexgroup interaction effect on the muscle (morphology) outcomes. Partial eta squared was reported for the main effects to provide a measure of magnitude. Post hoc pairwise comparisons identified differences between baseline and follow-up within each group. Analyses were performed using IBM SPSS (version 29, SPSS Inc., Chicago, IL, USA).

Finally, we qualitatively explored the muscle growth from baseline to one year post-BoNT-A in the intervention group in comparison to an age-matched cross-sectional SCP reference database. All graphs were designed in GraphPad Prism Software (Version 10, San Diego, CA, USA).

## Figures and Tables

**Figure 1 toxins-17-00327-f001:**
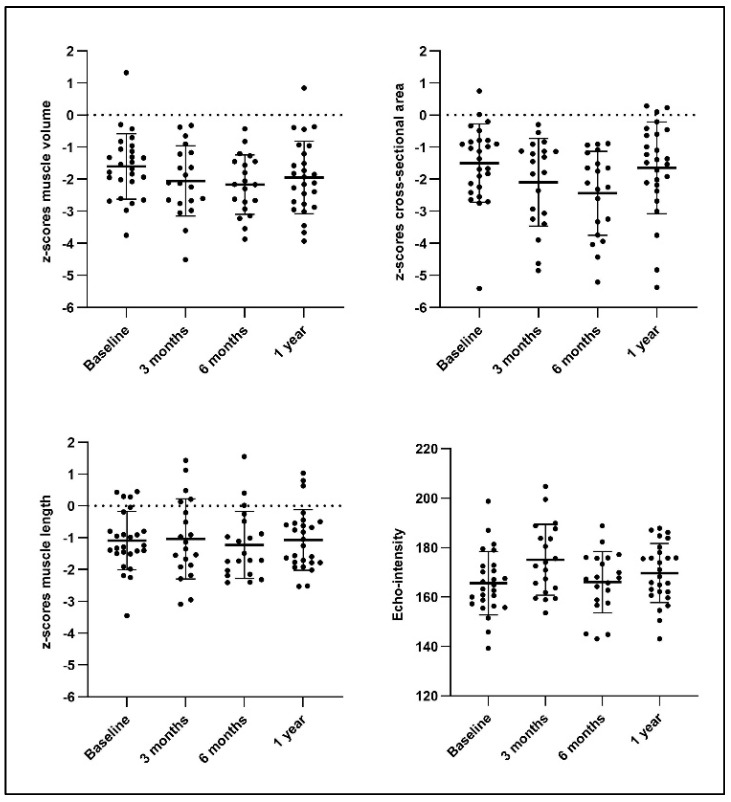
Box plots of individual datapoints at baseline, 3 months, 6 months, and 1 year post-botulinum neurotoxin type A (BoNT-A), with the means and standard deviations displayed as black stripes.

**Figure 2 toxins-17-00327-f002:**
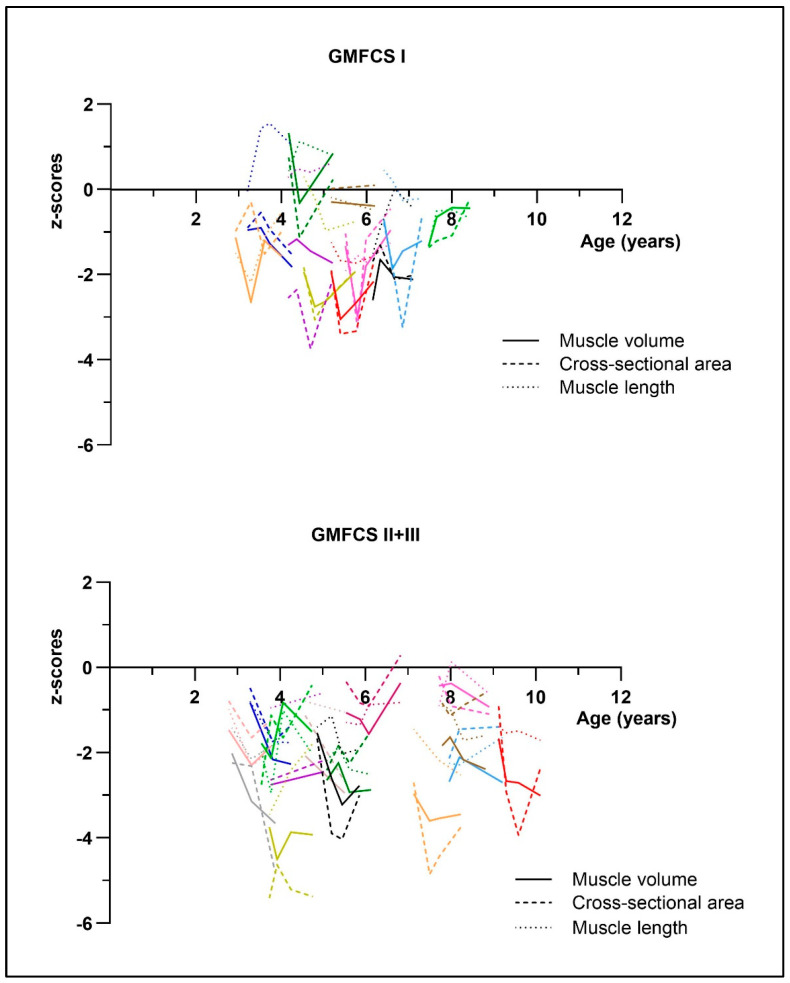
Overview of the individual trajectories of muscle morphology parameters per Gross Motor Function Classification System (GMFCS) level group. Each color represents an individual child.

**Figure 3 toxins-17-00327-f003:**
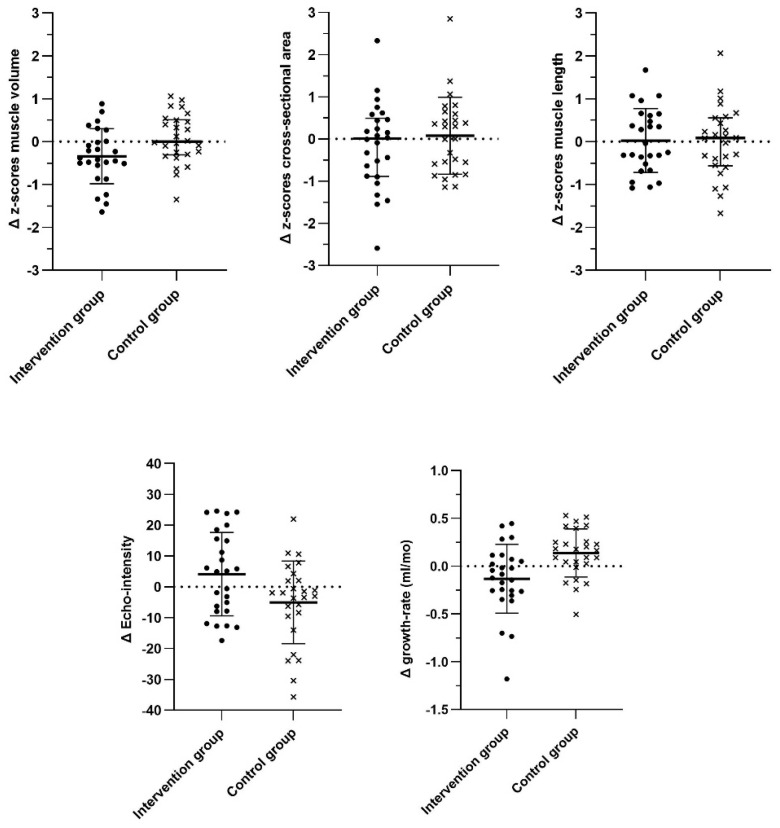
Box plots of the individual mean difference scores between baseline and 1 year post-botulinum neurotoxin type A for the intervention and control groups, with the mean and standard deviations displayed as black stripes. Positive mean difference values for the muscle morphology parameters indicate an increase in the absolute muscle outcome (i.e., decreased deficit) at the follow-up measurement. Positive mean difference values for echo intensity indicate an increase in echo intensity at the follow-up measurement and positive growth rate values indicate an increase in growth rate at the follow-up measurement. Note: ml, mililiter; mo, month.

**Figure 4 toxins-17-00327-f004:**
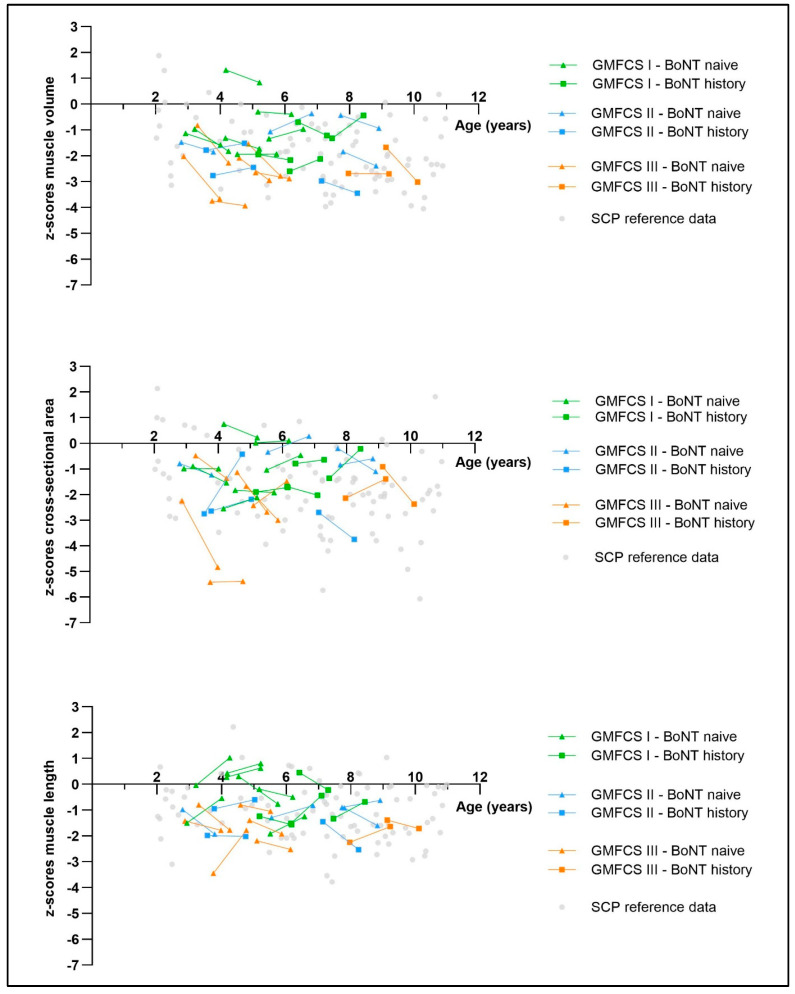
Scatterplots of individual baseline and 1-year follow-up data of the intervention group plotted against the spastic cerebral palsy (SCP) reference database. Triangles represent botulinum neurotoxin type A (BoNT-A)-naive children, and squares represent children with a BoNT-A history. The green color indicates children with a Gross Motor Function Classification System (GMFCS) level I, the blue color indicates children with GMFCS level II, and the orange color indicates children with GMFCS level III. The SCP reference data are displayed as grey dots. GMFCS, Gross Motor Function Classification System; BoNT-A, botulinum neurotoxin type A; SCP, spastic cerebral palsy; zMV, z-scores of muscle volume; zCSA, z-scores of the cross-sectional area; zML, z-scores of muscle length.

**Figure 5 toxins-17-00327-f005:**
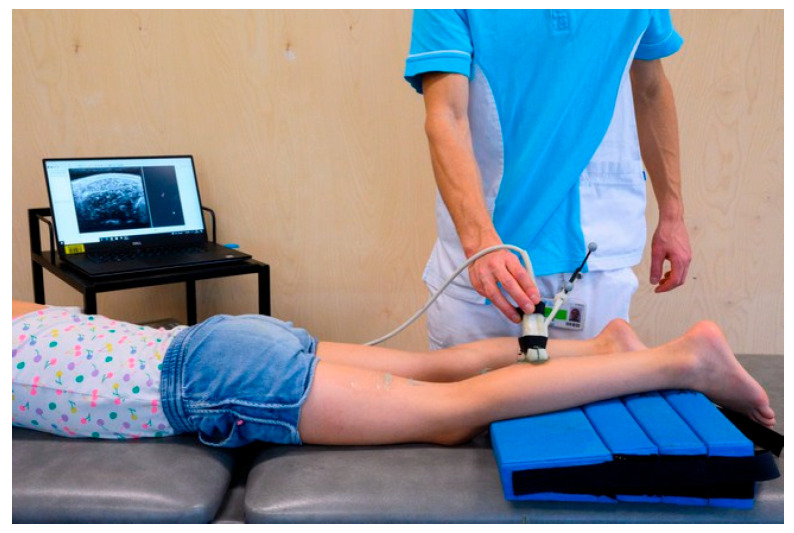
Illustration of the ultrasound set-up, including the motion tracking system (OptiTrack) and the B-mode ultrasound unit with a linear transducer (Telemed Echoblaster), both connected to a laptop for synchronized image and motion data acquisition and real-time display.

**Table 1 toxins-17-00327-t001:** Overview of the baseline patient characteristics of the intervention group, the control group, and the SCP reference group.

	Intervention Group	Control Group	SCP reference Group
Sample size	n = 26	n = 26	n = 87
Gender	female n = 10/male n = 16	female n = 14/male n = 12	female n = 31/male n = 56
Age (years)	5.19 (3.26)	4.98 (2.15)	6.96 (2.61)
Height (m)	1.05 (0.13)	1.06 (0.15)	1.19 (0.17)
Body weight (kg)	17.97 (5.70)	18.11 (5.35)	23.61 (7.99)
GMFCS level	I n = 11/II n = 7/III n = 8	I n = 11/II n = 7/III n = 8	I n = 55/II n = 21/III n = 11
Topographical involvement	unilateral n = 10 bilateral n = 16	unilateral n = 11 bilateral n = 15	unilateral n = 46 bilateral n = 41
BoNT-A-naive/history	naive n = 17 / history n = 9	naive n = 20 / history n = 6	naive n = 53 / history n = 34
Other injected muscles	hamstrings, n = 21 soleus, n = 6 psoas, n = 8 adductors, n = 6 rectus femoris, n = 2	NA	NA
AFO use post-BoNT-A	n = 26	NA	NA
Serial casting post-BoNT-A	n = 24	NA	NA

Data are presented as the means (standard deviations). SCP, spastic cerebral palsy; n, number; m, meter; kg, kilogram; GMFCS, Gross Motor Function Classification System; BoNT-A, botulinum neurotoxin type A; AFO, ankle foot orthosis; NA, not applicable.

**Table 2 toxins-17-00327-t002:** Pairwise differences between 4 different timepoints for muscle (morphology) outcomes of the intervention group.

Differences	zMV		zCSA		zML		EI	
	MD	*p*-Value	MD	*p*-Value	MD	*p*-Value	MD	*p*-Value
3 months−baseline	−0.50	0.005	−0.59	0.020	0.07	0.60	9.29	<0.0001 *
6 months−baseline	−0.42	0.003	−0.82	0.0005 *	0.01	0.95	−0.76	0.74
1 year−baseline	−0.34	0.008	−0.15	0.45	0.02	0.86	4.14	0.12
6 months−3 months	0.79	0.60	−0.22	0.27	−0.06	0.71	−10.05	0.004
1 year−3 months	0.16	0.36	0.43	0.047	−0.04	0.76	−5.15	0.12
1 year−6 months	0.08	0.44	0.655	0.003	0.02	0.86	4.90	0.032

Significant results at the 0.013 alpha level are indicated in bold to correct for multiple muscle parameters (alpha/4). Asterisks (*) indicate significant results at the 0.002 corrected alpha level for 6 pairwise comparisons. Mean differences of the estimates were calculated as follows: z-score at a later timepoint—z-score at an earlier timepoint. Positive mean difference values for the muscle morphology parameters indicate an increase in the absolute muscle outcome (i.e., decreased deficit) at the follow-up measurement. Positive mean difference values for echo intensity indicate an increase in echo intensity at the follow-up measurement. MD, mean difference; zMV, z-scores of muscle volume; zCSA, z-scores of the cross-sectional area; zML, z-scores of muscle length; EI, echo intensity.

**Table 3 toxins-17-00327-t003:** Pairwise differences between 4 different timepoints for muscle (morphology) outcomes for the GMFCS I and GMFCS II + III groups.

Differences	zMV		zCSA		zML		EI	
	MD	*p*-Value	MD	*p*-Value	MD	*p*-Value	MD	*p*-Value
*GMFCS I (n = 11)*								
3 months−baseline	−0.62	0.042	−0.64	0.056	0.20	0.35	6.82	0.06
6 months−baseline	−0.28	0.14	−0.88	0.004	0.22	0.43	−0.98	0.81
1 year−baseline	−0.12	0.44	0.09	0.55	0.25	0.26	3.82	0.33
6 months−3 months	0.34	0.087	−0.24	0.48	0.02	0.91	−7.80	0.14
1 year−3 months	0.50	0.076	0.73	0.053	0.06	0.78	−3.00	0.51
1 year−6 months	0.16	0.29	0.97	0.002 *	0.04	0.76	4.80	0.22
*GMFCS II + III (n = 15)*								
3 months−baseline	−0.37	0.043	−0.54	0.13	0.13	0.36	11.56	0.0001 *
6 months−baseline	−0.55	0.007	−0.76	0.024	−0.29	0.16	−0.67	0.79
1 year−baseline	−0.51	0.006	−0.32	0.31	−0.14	0.46	4.36	0.22
6 months−3 months	−0.19	0.38	−0.21	0.24	−0.41	0.16	−12.23	0.008
1 year−3 months	−0.14	0.42	0.22	0.33	−0.27	0.21	−7.20	0.12
1 year−6 months	0.04	0.76	0.43	0.14	0.14	0.31	5.03	0.066

Significant results at the 0.013 alpha level are marked in bold. Asterisks (*) indicate significant results at the 0.002 corrected alpha level for 6 pairwise comparisons. Mean differences of the estimates were calculated as follows: z-score at a later timepoint—z-score at an earlier timepoint. Positive mean difference values for the muscle morphology parameters indicate an increase in the absolute muscle outcome (i.e., decreased deficit) at the follow-up measurement. Positive mean difference values for echo intensity indicate an increase in echo intensity at the follow-up measurement. MD, mean difference; zMV, z-scores of muscle volume; zCSA, z-scores of the cross-sectional area; zML, z-scores of muscle length; EI, echo intensity.

**Table 4 toxins-17-00327-t004:** Pairwise differences between 4 different timepoints for muscle (morphology) outcomes for the botulinum neurotoxin type A (BoNT-A)-naive and BoNT-A history groups.

Differences	zMV		zCSA		zML		EI	
	MD	*p*-Value	MD	*p*-Value	MD	*p*-Value	MD	*p*-Value
*BoNT-A-naive (n = 17)*								
3 months−baseline	−0.65	0.004	−0.66	0.031	0.17	0.29	10.09	0.002 *
6 months−baseline	−0.63	<0.0001 *	−0.69	0.0009 *	−0.11	0.59	−2.55	0.28
1 year−baseline	−0.49	0.002 *	−0.35	0.10	0.02	0.90	5.70	0.098
6 months−3 months	0.02	0.93	−0.02	0.93	−0.27	0.092	−12.64	0.001 *
1 year−3 months	0.17	0.52	0.31	0.33	−0.15	0.46	−4.39	0.28
1 year−6 months	0.15	0.28	0.33	0.19	0.13	0.13	8.25	0.002 *
*BoNT-A history (n = 9)*								
3 months−baseline	−0.22	0.43	−0.44	0.33	−0.07	0.72	8.40	0.012
6 months−baseline	−0.01	0.96	−0.99	0.052	0.35	0.25	4.24	0.42
1 year−baseline	−0.07	0.74	0.24	0.51	0.02	0.90	1.17	0.76
6 months−3 months	0.21	0.28	−0.55	0.037	0.42	0.16	−4.16	0.49
1 year−3 months	0.15	0.41	0.68	0.011	0.09	0.58	−7.23	0.19
1 year−6 months	−0.06	0.70	1.22	0.0002 *	−0.32	0.034	−3.07	0.43

Significant results at the 0.013 alpha level are marked in bold. Asterisks (*) indicate significant results at the 0.002 corrected alpha level for 6 pairwise comparisons. Mean differences of the estimates were calculated as follows: z-score at a later timepoint—z-score at an earlier timepoint. Positive mean difference values for the muscle morphology parameters indicate an increase in the absolute muscle outcome (i.e., decreased deficit) at the follow-up measurement. Positive mean difference values for echo intensity indicate an increase in echo intensity at the follow-up measurement. MD, mean difference; zMV, z-scores of muscle volume; zCSA, z-scores of the cross-sectional area; zML, z-scores of muscle length; EI, echo intensity.

**Table 5 toxins-17-00327-t005:** Pairwise differences of muscle (morphology) outcomes between baseline and follow-up for the intervention group and the control group.

	Intervention group				Control group			
	Baseline	Follow-Up	MD/%	*p*-Value	Baseline	Follow-Up	MD/%	*p*-Value
zMV	−1.61 (−2.050, −1.16)	−1.95 (−2.46, −1.43)	−0.34	0.007	−1.83 (−2.27, −1.38)	−1.75 (−2.27, −1.24)	0.08	0.55
zCSA	−1.50 (−2.052, −0.94)	−1.65 (−2.28, −1.010)	−0.15	0.44	−1.61 (−2.17, −1.057)	−1.54 (−2.17, −0.90)	0.07	0.69
zML	−1.09 (−1.51, −0.68)	−1.07 (−1.48, −0.66)	0.02	0.87	−1.01 (−1.43, −0.60)	−1.02 (−1.43, −0.61)	−0.01	0.98
EI	165.61 (160.36, 170.86)	169.75 (165.060, 174.43)	4.14	0.12	158.86 (153.61, 149.15)	153.84 (149.15, 158.52)	−5.02	0.063
Growth rate (mL/mo)	0.44 (0.39, 0.49)	0.31 (0.18, 0.44)	−0.13/ −30%	0.035	0.44 (0.39, 0.49)	0.58 (0.45, 0.71)	0.14/ +32%	0.029

Overview of the estimates (means) with the confidence interval (CI, lower bound, upper bound) for muscle (morphology) parameters per group. Muscle growth rate was calculated as the ratio of muscle volume (milliliters) per age (months). The growth rate at follow-up was calculated as the ratio of the change in muscle volume (mL) to the time of follow-up (months). Significant results at the 0.013 alpha level are marked in bold to correct for 4 muscle outcomes. No Bonferroni correction was applied for growth rate. Mean differences of the estimates were calculated as follows: z-score at 1-year follow-up—z-score at baseline. Positive mean difference values for the muscle morphology parameters indicate an increase in the absolute muscle outcome (i.e., decreased deficit) at the follow-up measurement. Positive mean difference values for echo intensity indicate an increase in echo intensity at the follow-up measurement. MD, mean difference; zMV, z-scores of muscle volume; zCSA, z-scores of the cross-sectional area; zML, z-scores of muscle length; EI, echo intensity; ml, mililiter; mo, month.

## Data Availability

All raw data supporting the conclusions of this article will be provided by the authors upon request, with no unwarranted limitations.

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
