# Peer review of "Botulinum Neurotoxin A-Induced Muscle Morphology Changes in Children with Cerebral Palsy: A One-Year Follow-Up Study"

_toxins, 2025, doi:10.3390/toxins17070327_

Round 1
Reviewer 1 Report
Comments and Suggestions for Authors
This study represents a valuable contribution to the field, using a robust methodology to address an important clinical question.
The manuscript presents comprehensive ultrasound data on muscle morphology in children with spastic cerebral palsy post-BoNT-A injection. However, it would benefit from including a brief discussion highlighting that spastic muscles already exhibit distinct ultrasound characteristics compared to typically developing muscles. For example, spastic muscles may demonstrate increased echogenicity, disrupted intramuscular fascia, fibrosis, atrophy, sarcomere loss, and fatty infiltration, all of which impact ultrasound imaging and may influence BoNT-A injection strategies and recovery trajectories.
Please see this paper: Popescu MN, CăpeÈ› C, Beiu C, Berteanu M. The Elias University Hospital Approach: A Visual Guide to Ultrasound-Guided Botulinum Toxin Injection in Spasticity: Part I—Distal Upper Limb Muscles. Toxins. 2025; 17(3):107. https://doi.org/10.3390/toxins17030107
Reviewer 2 Report
Comments and Suggestions for Authors
Dear Authors,
Please follow my suggestions to improve the manuscript structures and contents.
1. Please add the keywords.
2. ABSTRACT and METHODS:
-
Indicate when the botulinum toxin is repeated. If it is administered again after one year, please explain the rationale, as it is well known that the effects do not typically last that long.
-
Specify whether the ultrasound (US) was performed by multiple operators or by the same clinician throughout. This is essential to assess the repeatability and reliability of the results.
- Please specify clearly the design of the study in the title, abstract and methods.
3. INTRODUCTION:
Make the introduction more concise and fluid, avoiding overly general or well-known concepts. Aim to reduce the length to a maximum of one and a half pages. Additionally, consider that comparisons with other studies are more appropriate for the Discussion section.
4. RESULTS:
Include a figure illustrating the measurement method used with US.
5. DISCUSSION:
Emphasize the strengths of the study, particularly the potential for repeatability of the ultrasound measurements. Highlight that simple, reproducible, and non-invasive ultrasound assessments of muscle structure can clearly demonstrate differences in muscle morphology (DOI: 10.1590/1806-9282.65.2.165).
6. MAIN TEXT:
The main limitations of the study are its organizational structure, the small sample size and the repeatibility of US measurements. Consider restructuring the manuscript to include a clearer and more concise Introduction, and use the Discussion to underline comparisons with similar studies, highlighting the relevance and originality of your findings.
